# Advancing Cancer Research: Current Knowledge on Cutaneous Neoplasia

**DOI:** 10.3390/ijms241311176

**Published:** 2023-07-06

**Authors:** Laura Stătescu, Laura Mihaela Trandafir, Elena Țarcă, Mihaela Moscalu, Maria Magdalena Leon Constantin, Lăcrămioara Ionela Butnariu, Mioara Florentina Trandafirescu, Mihaela Camelia Tîrnovanu, Rodica Heredea, Andrei Valentin Pătrașcu, Doru Botezat, Elena Cojocaru

**Affiliations:** 1Medical III Department, Faculty of Medicine, “Grigore T. Popa” University of Medicine and Pharmacy, 700115 Iasi, Romania; laura.statescu@umfiasi.ro; 2Department of Mother and Child, Faculty of Medicine, “Grigore T. Popa” University of Medicine and Pharmacy, 700115 Iasi, Romania; laura.trandafir@umfiasi.ro (L.M.T.); ionela.butnariu@umfiasi.ro (L.I.B.); 3Department of Surgery II—Pediatric Surgery, “Grigore T. Popa” University of Medicine and Pharmacy, 700115 Iasi, Romania; 4Department of Preventive Medicine and Interdisciplinarity, “Grigore T. Popa” University of Medicine and Pharmacy, 16 University Street, 700115 Iasi, Romania; moscalu.mihaela@gmail.com (M.M.); doru.botezat@umfiasi.ro (D.B.); 5Medical I Department, Faculty of Medicine, “Grigore T. Popa” University of Medicine and Pharmacy, 700115 Iasi, Romania; leon_mariamagdalena@yahoo.com; 6Department of Morphofunctional Sciences I–Pathology, “Grigore T. Popa” University of Medicine and Pharmacy, 700115 Iași, Romania; mioaratrandafirescu@yahoo.co.uk (M.F.T.); andrei-valentin-a-patrascu@email.umfiasi.ro (A.V.P.); elena2.cojocaru@umfiasi.ro (E.C.); 7Department of Clinical Practical Skills, “Victor Babeş” University of Medicine and Pharmacy, 300041 Timisoara, Romania; elena-rodica.heredea@umft.ro

**Keywords:** malignant melanoma, non-melanocytic skin cancer, cutaneous lymphoma

## Abstract

Skin cancers require a multidisciplinary approach. The updated guidelines introduce new insights into the management of these diseases. Melanoma (MM), the third most common skin cancer, a malignant melanocytic tumor, which is classified into four major histological subtypes, continues to have the potential to be a lethal disease. The mortality–incidence ratio is higher in Eastern European countries compared to Western European countries, which shows the need for better prevention and early detection in Eastern European countries. Basal cell carcinoma (BCC) and squamous cell carcinoma (cSCC) remain the top two skin cancers, and their incidence continues to grow. The gold standard in establishing the diagnosis and establishing the histopathological subtype in BCC and SCC is a skin biopsy. Sebaceous carcinoma (SeC) is an uncommon and potentially aggressive cutaneous malignancy showing sebaceous differentiation. It accounts for 0.7% of skin cancers and 3–6.7% of cancer-related deaths. Due to the rapid extension to the regional lymph nodes, SeC requires early treatment. The main treatment for sebaceous carcinoma is surgical treatment, including Mohs micrographic surgery, which has the advantage of complete margin evaluation and low recurrence rates. Primary cutaneous lymphomas (PCLs) are a heterogeneous group of lymphoproliferative diseases, with no evidence of extracutaneous determination at the moment of the diagnosis. PCLs have usually a very different evolution, prognosis, and treatment compared to the lymphomas that may secondarily involve the skin. The aim of our review is to summarize the important changes in the approach to treating melanoma, non-melanoma skin, cutaneous T and B cell lymphomas, and other types of skin cancers. For all skin cancers, optimal patient management requires a multidisciplinary approach including dermatology, medical oncology, and radiation oncology.

## 1. Introduction

In recent years dermato-oncology has had important developments due to genetic studies, new classifications and staging of diseases, but also due to new therapeutic agents. Malignant melanoma remains in the attention of dermatologists and oncologists due to its potential lethal evolution. In recent years, melanoma has become one of the most common cancers and the trend is continuously growing in regard to the number of cases [1,2]. Suspicious pigmented lesions need to be evaluated, and diagnosis must be based on clinical examination, dermoscopy, and full-thickness excisional biopsy, with a minimal side margin, according to international guidelines. Staging and risk assessment procedures are defined by the characteristics of the disease presentation at diagnosis [2,3,4]. Non-melanoma skin cancers (basal cell carcinoma—BCC and squamous cell carcinoma—SCC) remain in the first two places in regard to skin cancers, but the impact on the survival rate is not so high in most cases. A small percentage of cases involving locally advanced SCC and metastatic SCC can significantly influence the overall survival rate. Starting from 2019, there have been systemic therapies approved for these types of SCC [5]. Primary cutaneous lymphomas (PCLs), represented by a heterogenous group of lymphoproliferative diseases, require a multidisciplinary approach, and the latest classification of WHO–EORTC (published in 2019) includes new distinct types of lymphomas [6].

The aim of our review is to summarize the important changes in the approach to treating melanoma, non-melanoma skin, and cutaneous T and B cell lymphomas.

## 2. Risk Factors, Causes, and Major Types of Skin Cancer and Their Treatments

### 2.1. Malignant Melanoma

#### 2.1.1. Cutaneous Melanoma (CM)

Melanoma, the third most common skin cancer, is a malignant melanocytic tumor, classified into four major subtypes: superficial spreading, nodular, acral lentiginous, and lentigo maligna melanomas [1]. It continues to have the potential to be a lethal disease. In the United States, there has been a persistent increase in the incidence of cutaneous neoplasia since 1975, with 91,320 new cases diagnosed in 2018 alone. Furthermore, in the US, cutaneous melanoma incidence rates have continued to increase among the younger age groups (15–29-year-olds, and mostly in the 20–29-year-old age groups) and melanoma is the fourth most frequently diagnosed invasive cancer in this age group [2,3].

The annual incidence of MM in Europe varies from 3.5/100,000 in the Mediterranean countries to 12–35/100,000 in the Nordic countries [1]. Australia and New Zealand have an annual incidence of over 50/100,000. In Australia, cutaneous neoplasia ranks as the eighth leading cause of cancer-related fatality. The mortality–incidence ratio is higher in Eastern European countries compared to Western European countries, which shows the need for better prevention and early detection in Eastern European countries [4].

CM is initially diagnosed through a visual inspection of the skin. Diagnostic cues include asymmetry, border, color, diameter evolution, and the “ugly duckling” sign (moles that are different or stand out in comparison to other moles on one’s body) [7]. Importantly, the use of dermoscopy by experienced people can improve the accuracy of the diagnosis. Moreover, there are various emerging technologies aiming to enhance the precision of pre-biopsy diagnosis. These advancements encompass artificial intelligence-driven image analysis, three-dimensional imaging of the entire body, reflectance confocal microscopy (RCM), optical coherence tomography (OCT), and the extraction of epidermal genetic information through adhesive tape stripping [2].

Despite their promising potential, none of these technologies, except for digital photographic monitoring, are currently employed in clinical practice. The number of cutaneous neoplasia cases detected through whole body photography and sequential digital dermoscopy is still lower compared to those identified through conventional methods [4]. As technology becomes lighter and more accessible, RCM and OCT are likely to be used more and more in the future [2]. Diagnosis is based on narrow-margin excision biopsy and the histology report should follow the eighth edition of the American Joint Committee on Cancer (AJCC) tumor, node, metastasis (TNM) classification, and include information such as the maximum thickness in millimeters (Breslow), presence of ulceration, clearance of surgical edges, and also the assessment and recording of mitotic rate and regression because they are particularly important [7]. Immunohistochemistry helps to diagnose melanoma, especially amelanotic and extracutaneous lesions [8].

There are novel diagnostic approaches for skin cancer that eliminate the need for invasive skin sample removal, but a biopsy with histopathological examination remains the gold standard for diagnosing melanoma. One such method is known as RCM, which enables doctors to examine abnormal skin areas to a specific depth without the need for cutting into the skin. RCM is utilized in Europe and in select centers across the United States [2]. Its application can be particularly valuable for individuals with numerous atypical moles, as it reduces the necessity for multiple skin biopsies. Additionally, RCM shows promise in delineating the borders of a melanoma, aiding surgeons during the surgical procedure. As this technique continues to evolve, it is expected to become more widely available in the coming years [2].

Currently under investigation is a technique called adhesive patch testing, which offers an alternative to traditional skin biopsy. Instead of cutting into the skin, a sticky patch is applied to the suspicious area. Upon removal, the patch collects the upper layers of skin, which can then be subjected to testing for specific gene alterations frequently associated with melanoma. If any of these gene changes are detected, a standard biopsy of the area can be performed. In cases where no gene changes are found, a biopsy is unnecessary, and the area can be monitored instead.

While most early-stage melanomas can be effectively cured through surgical intervention, a small proportion of these cancers eventually metastasize to other parts of the body, making treatment challenging. Recent research has revealed that certain gene expression patterns within melanoma cells can provide insights into the likelihood of spread for stage I or II melanomas. Building upon this research, a laboratory test known as DecisionDx-Melanoma has been developed and is currently available and is based on the biologic profile of 31 genes within the tumor tissue, according with the American Cancer Society [7]. This test classifies melanomas into two primary groups based on their gene patterns: class 1 tumors exhibit a reduced risk of metastasis and class 2 tumors carry an increased risk of metastasis. This test could assist in determining whether individuals with early-stage melanoma should undergo a sentinel lymph node biopsy (SLNB), consider additional treatment, or receive closer monitoring following treatment to detect potential recurrence [9].

Malignant melanoma originates from atypical and genetically mutated melanocytes. Congenital melanocytic nevi (CMN) have an increased risk of developing into cutaneous melanomas; the risk of developing melanomas in childhood and adolescence is increased by 465 times. The risk is higher in giant CMN. BRAF mutations are frequently observed in association with acquired nevi, similar to the presence of these mutations in giant congenital melanocytic nevi (CMN) associated with neurocutaneous syndrome, found in approximately 80% of cases [7]. Lesions of the Spitzoid type are classified into three categories: Spitz nevus, atypical Spitz nevus, and Spitzoid melanoma.

Malignant tumors consistently exhibit positive staining for S-100, HMB-45, and Melan-A, while benign tumors show p-16 staining.

Genetically benign lesions commonly exhibit HRAS mutations, atypical lesions display chromosomal abnormalities such as BAP1 loss, and melanomas often present NRAS and BRAF mutations. Genetic analyses of histologic subtypes reveal a higher prevalence of BRAF mutations in superficial spreading melanoma (SSM), while NRAS mutations are more frequently observed in nodular melanoma (NM) [10]. Some people inherit mutant genes from their parents that increase their risk of malignant melanoma. For example, changes in the CDKN2A gene (p16) cause some melanomas to occur in certain families [8]. Molecular characterization is an essential requirement for patients with stage III or IV (I, A) melanoma, and it is highly recommended for those with high-risk resected disease in stage IIC, but not for patients in stage I or stages IIa-IIb [1]. Testing for BRAF mutations is mandatory (I, A).

The primary melanoma subtypes that are commonly linked to specific gene mutations are as follows:BRAF/CDKN2A/NRAS/TP53—cutaneous melanoma;BRAF/NRAS/NF1/KIT—acral melanoma;SF3B1—mucosal melanoma [11].

There are other indices such as programmed death-ligand-1 (PD-L1) expression, reported as the percentage of positive tumor cells, that can be useful for assessing and recording all resectable or unresectable stage III and IV [I, B] melanomas; however, its clinical use is very limited at this time and is not currently warranted [2,12]. The eighth version of the AJCC staging and classification system, which includes sentinel lymph node (SN) staging, is currently the preferred classification system, and brings changes in the tumor staging, nodal staging, and metastasis staging, as follows.

#### 2.1.2. Changes in Tumor Staging

One major change in tumor staging is the redefinition of T1a as less than 0.8 mm and T1b as greater than 0.8 mm or less than 0.8 mm but with ulceration. In the previous seventh edition, the transition point was set at 1 mm. The inclusion of tumor thickness proved to have more prognostic significance than the presence or absence of a mitotic figure, leading to the removal of mitotic rate as a defining factor [13]. However, recording the mitotic rate is still important as it holds predictive value, which will likely be further understood in future studies. The significance of determining T1a/b status lies in the distinction between a less than 5% likelihood of sentinel lymph node (SLN) metastases for T1a versus 5% to 12% for T1b, making SLNB recommended for T1b cases [2].

#### 2.1.3. Changes in Nodal Staging

A significant change in nodal staging for melanoma is the incorporation of tumor status into the overall staging. In the previous seventh edition, the presence of nodal status took precedence over the associated tumor stage. The inclusion of tumor thickness and ulceration results in a more detailed distribution of stage III, with three to four categories (stages IIIA, IIIB, IIIC, and IIID). Terms like microscopic and macroscopic metastases are now referred to as clinically occult if detected by SLNB, and clinically evident if identified through clinical or radiographic examination. Additionally, the presence of microsatellites, satellites, or in-transit metastases automatically falls under N1c, N2c, and N3c, respectively, depending on the number of tumor-involved lymph nodes (1, 2 or 3, and 4 or more) [2]. Microsatellites are defined as microscopic cutaneous or subcutaneous tumors discontiguous from the primary site but identified during the pathological examination of the original primary site.

#### 2.1.4. Changes in Metastases Staging

The major changes in the staging of metastasis status in melanoma include the addition of central nervous system (CNS) metastases as a distinct class, Md, and the inclusion of lactate dehydrogenase (LDH) as an additional prognostic factor for each M stage [2].

#### 2.1.5. Mutational Subtypes in Melanoma

Exomic sequencing is routinely performed on metastatic melanoma samples to identify key mutations that have an impact on treatment and prognosis. The Cancer Genome Atlas Network has recently proposed a classification schema for cutaneous melanoma (CM) based on mutations, including BRAF, RAS, NF-1, and triple-wild type (WT) [14]. It is anticipated that genomic classification will increasingly replace the current pathological subtyping system used for superficial spreading, nodular, acral lentiginous, and lentigo maligna melanomas. Assessing the mutational status of a patient’s tumor represents a move toward personalized medicine in terms of both prognosis and treatment. A notable example is the highly targeted approach of inhibiting the kinase domain of BRAF, which has shown significant success in halting the progression of BRAFV600 mutant metastatic melanoma [15].

Management of Local/Locoregional Disease of MM is summarized in Table 1.

Summary of recommendations in the adjuvant settings, based on clinical trials [1]:PD-1 blockade dabrafenib/trametinib—stage IIIA, B, C (SN > 1 MM) for BRAF-mutated melanoma;Combination nivolumab + ipilimumab shows exceptionally encouraging clinical activity compared to nivolumab in stage IV patients shown by NED via surgery or radiotherapy, but with increased toxicity [19];BRAF WT patients—PD-1 blockade is the only advised alternative;BRAF-mutated melanoma—the decision is dependent on the toxicity profiles.

### 2.2. Non-Melanocytic Skin Cancer

#### 2.2.1. Basal Cell Carcinoma (BCC)

BCC exhibits a continuous increase in incidence and is recognized as the most prevalent malignant tumor in White (European descent) populations, constituting 80–85% of all skin cancers [22,23,24,25,26]. The incidence of BCC is steadily increasing regardless of geographic area (the highest incidence was reported in Australia, followed by the United States of America and Europe). In Europe, the average incidence of BCC is 76.21/100,000 people [27]. Statistics for the global level show that the incidence increases by 10% annually, which can lead to doubling the incidence rate in the next 30 years [28]. The risk factors of BCC include ultraviolet radiation A and B (UVA and UVB; the most important carcinogenetic risk factor in individuals with light-colored skin and hair, with low tanning capacity who are frequently exposed to the sun, especially older men), ionizing radiation, arsenic ingestion, and immune suppression in organ transplantation [23,26,28]. Newly-formed pigmented nevi, sunburn, and the presence of actinic keratosis are associated with an increased risk of BCC and, they are correlated with exposure to UV [25]. BCC is commonly located in areas exposed to the sun but is also found in the genital and perianal region, in the palm, sole, and nails [22]. Standard histological subtypes are predisposed to certain anatomical areas and are characterized by certain clinical and histological features (Table 2). The typical clinical manifestation of BCC is papular with a pearly or telangiectatic appearance, or sometimes with ulcerations or erosions. BCC rarely metastasizes or causes death, but it can result in extensive morbidity through local invasion and tissue destruction [23,29].

The gold standard in establishing the diagnosis and establishing the histopathological subtype in BCC is a skin biopsy [22,36]. Dermoscopy plays a significant role in establishing the diagnosis of basal cell carcinoma (BCC) with a high accuracy range of 95–99%. It aids in distinguishing BCC from other neoplastic and inflammatory conditions, and in differentiating between BCC subtypes before resorting to skin biopsy. Reflectance confocal microscopy (RCM) and optical coherence tomography (OCT) are emerging noninvasive diagnostic techniques that offer valuable insights in BCC diagnosis [29]. Combining RCM with dermoscopy allows the identification of BCC subtypes without skin biopsy, but with certain limitations regarding depth and edges when interpreting images. Non-invasive real-time diagnosis involved in BCC management includes OCT, which involves infrared light on the skin. This method if combined with dermoscopy can reach a higher sensitivity and specificity (diagnostic accuracy of 87.4%) [37].

Multiple tumor suppressor genes and proto-oncogenes play a role in the development of basal cell carcinoma (BCC). These include key components of the Hedgehog pathway such as Patched Homolog 1 (PTCH1) and smoothened (SMO), as well as the tumor protein p53 (TP53) and members of the RAS proto-oncogene family [28]. Mutations of the PTCH1 gene on chromosome 9 occur in between 30 and 90% of patients diagnosed with BCC, and in patients with basal cell nevus syndrome (BCNS) or Gorlin syndrome. BCNS is an autosomal dominant disorder characterized by multiple BCCs starting at a young age, and which can develop into other tumors (e.g., medulloblastomas). The diagnosis must be established quickly because without treatment, BCC can invade the skin and deep tissues, such as muscle and bone. Recently, new drugs that inhibit targets in the Sonic Hedgehog pathway have been developed [38].

Other genetic abnormalities that predispose individuals to BCC as well to other visceral tumors include xeroderma pigmentosum, Basez–Dupré–Christol syndrome, Rombo syndrome, Basex syndrome, and Muir–Torre syndrome [23,39]. BCC tumors are characterized by a great clinico-histological heterogeneity based on distinct molecular subtypes, on which the aggressiveness and the response to treatment depend [36].

Although there are currently many therapeutic options for BCC, the “gold standard” is surgical excision through radical resection. This represents the most effective treatment method and allows for a confirmation of the histological diagnosis [25,27]. In the case of BCC with a risk for subclinical spread or local recurrence, standard surgical excision is recommended. The recommendations for safety margins in the standard excision of basal cell carcinoma (BCC) differ based on the risk profile of each individual tumor. For low-risk tumors, the recommended range for peripheral margins is between 2 mm and 5 mm. In contrast, high-risk lesions are advised to have peripheral margins between 5 mm and 15 mm [27,40]. In case the surgical resection did not lead to optimal results, or in the case of deep invasive tumors, radiotherapy is recommended [22]. When radical excision is not possible, one of the following therapeutic options is used:Destructive therapies such as curettage, electrocautery (electrodessication), cryotherapy, and laser ablation are viable treatment options for small, low-risk, non-facial basal cell carcinoma (BCC) as well as for multiple small BCCs [41].Topical therapies including imiquimod, 5-fluorouracil, and vismodegib are recommended for patients with low-risk BCC who prefer a non-surgical intervention, or for whom surgery is contraindicated due to patient-related factors such as age, comorbidities, or medications. Vismodegib is specifically used for patients with metastatic or advanced BCC and for some patients with Gorlin syndrome [42].Photodynamic therapy with 5-aminolevulinic acid or its methylester (methyl-5-amino-4-oxopentanoate) should be considered for patients with non-aggressive, low-risk BCC, particularly small superficial and nodular types with a tumor thickness not exceeding 2 mm, when surgery is unsuitable or contraindicated due to patient-related limitations [26,29,36,37]. Combining therapies is based on the principle that their mechanisms of action complement or enhance each other. Combination therapies may be considered for selected patients with BCC lesions where surgical outcomes may result in significant disfigurement or may have a low expected curative rate.

#### 2.2.2. Squamous Cell Carcinoma (SCC)

SCC ranks as the second most prevalent form of malignancy after BCC. Notably, SCC is responsible for most deaths associated with non-melanoma skin cancer [43]. It is a heterogeneous tumor which originates in the epidermal keratinocytes, showing different degrees of differentiation and cytological features depending on their resemblance to the cells from which they develop [44]. Surgical resection offers a favorable outcome for most of the patients [45]. Data regarding the incidence of non-melanoma skin cancer show that SCC and BCC are the most frequent cancers in Caucasians in the general population, being more common in men than in women, and frequently seen in individuals with a lighter skin pigmentation [46]. The incidence varies depending on the geographic area, between 5–499 per 1000 individuals [47]. Cancer archives worldwide often contain controversial and incomplete and/or underreported data on non-melanoma skin cancers; therefore, their real incidence is underestimated [48]. Recently, in Europe, it has been proposed that cancers initiated by identifiable risk factors that are responsive to preventive actions should be noted in Cancer Registries’ agendas [46,49]. The risk factors associated with the development of SCC in different populations vary a lot. Epidemiological studies from the latest decades show a decrease in occupational exposure and chemical carcinogens at the same time as an increase in cases triggered by sun exposure, solar radiation, and UV, being categorized among the group 1 carcinogens by the International Agency for Research on Cancer (IARC). Other factors involved in the etiology of SCC are actinic keratosis, chronic immunosuppression, chronic inflammation, topical carcinogens, additional forms of radiation, burn scars, human papillomavirus infection, arsenic, and xeroderma pigmentosa [44]. The most frequent anatomic sites for SCC include the lips, nose, eyelids, hairless scalp, ears, arms, trunk, and legs as they are the most constantly sun-exposed areas or are related to indoor tanning; however, lesions associated with HPV infection or immunosuppression, the genitalia, and the subungual region of distal digits are also frequently described [43].

The diagnosis of SCC is usually clinical, with subsequent histological validation following surgical excision. SCC in situ may have a hyperkeratotic lesion appearance requiring a differential diagnosis with benign keratosis, lichen simplex chronicus, or a dermatosis. The invasive SCC may show an ulcerated hyperkeratotic nodule, but the gross appearance varies with the degree of differentiation. For instance, less well differentiated SCC may look like thin or thick irregular, erythematous, prominent nodules and plaques showing induration, ulceration, and hemorrhage; keratoacanthoma, which is a well-differentiated SCC, gives rise to crateriform lesions having central keratin masses. The lymphatic pathway is the favorite route for metastasis, but visceral lesions may develop, commonly in the lung, bone, central nervous system, and liver [43,45,48]. Distinctive gross features are suggestive of the diagnosis, but a confident diagnosis is established via the histological examination of shave, punch, or excisional biopsies, even if immunohistological examination is required sometimes [49].

In order to distinguish SCC from other types of cancer, immunohistochemistry (IHC) is used as a valuable tool, confirming its squamous differentiation, and providing additional information about the tumor’s characteristics [50]. Specific markers are used to detect the expression of proteins that are typically present in SCC cells [51]. Some commonly used IHC markers in SCC include cytokeratins (such as CK5/6, CK7, CK14), p63, p40, and E-cadherin [52]. These markers help to identify the presence of squamous differentiation and differentiate SCC from other types of cancer, such as adenocarcinoma or small-cell carcinoma [51]. In addition to diagnosis, IHC can also provide information about the prognosis and can guide treatment decisions [50]. For example, markers like Ki-67, p53, and EGFR (epidermal growth factor receptor) can help to assess the tumor’s aggressiveness, proliferative activity, and potential response to targeted therapies [53]. IHC plays a crucial role in evaluating the expression of immune checkpoint markers such as PD-L1 (programmed cell death ligand 1) on tumor cells and infiltrating immune cells. This information helps to predict the response to immunotherapy and guide treatment decisions [54]. IHC can assess the composition and activity of the tumor microenvironment in SCC. It enables the characterization of immune cell populations, such as tumor-infiltrating lymphocytes, regulatory T cells, and myeloid-derived suppressor cells. Understanding the immune landscape of SCC can provide insights into the tumor’s immune evasion mechanisms [53]. Ongoing research has identified novel IHC markers that can serve as predictive and prognostic indicators in SCC. For example, markers like SOX2 (sex-determining region Y-box 2) and p16INK4a (cyclin-dependent kinase inhibitor 2A) have shown promise in predicting treatment response and prognosis in SCC of the head and neck region [55,56].

The future of IHC in SCC diagnosis and management is promising. Advancements in technology and the discovery of novel markers are continuously improving their utility. Moreover, the integration of IHC with molecular testing, such as genetic profiling and next-generation sequencing, further enhances the characterization of SCC and facilitates personalized treatment strategies [54].

Generally, SCC begins from in situ lesions with full-thickness keratinocytic atypia which develop progressively and give rise to invasive forms, varying from well to moderately and poorly differentiated lesions. A small number of lesions have no noticeable in situ precursor. The prognosis is correlated with the staging of SCC which takes into account lesion size, invasion depth, differentiation, and perineural invasion. For poorly differentiated forms, immunohistochemistry is typically positive for p63, p40, EMA, CK5/6, MNF116, and high-molecular-weight 34ßE12, and negative for CAM 5.2, CK20, S100, SOX10, MelanA, and BerEP4 [44]. High-risk prognostic factors are tumor thickness >2 mm, Clark level IV or V invasion, poor differentiation, primary site on the lip or ear, and perineural invasion [47,49].

SCC differential diagnoses include inflammatory dermatoses, various keratoses, or other malignant tumors. Well-differentiated SCC, for instance, must be differentiated from pseudo-epitheliomatous hyperplasia, syringometaplasia, perineural hyperplasia, warts, endophytic keratosis, and adnexal tumors with a squamous appearance; poorly differentiated SCC need to be distinguished from melanoma, sarcoma, atypical fibroxanthoma, and lymphoma, requiring immunohistochemical examination [44]. SCC seems to progress through a multistep process. The pathogenesis of SCC related to UV radiation develops via mutations which involve tumor suppressor genes, such as *TP53*, *CDKN2A*, *NOTCH1*, and *NOTCH2*, *TERT*, *EGFR*, or activating mutations in genes such as HRAS and KRAS. Molecular pathways such as RAS/RAF/MEK/ERK and PI3K/AKT/mTOR play a significant role in the pathophysiology of this tumor [57].

Most SCCs are clonal in origin; therefore, they develop from the same single cell, probably from an epidermal or follicular stem cell [58]. In recent years, genetic research has made significant progress in uncovering the genetic factors associated with cutaneous squamous cell carcinoma (SCC). Through these studies, 14 single-nucleotide polymorphisms (SNPs) related to SCC have been identified [59,60]. A meta-analysis conducted by Sarin et al. in 2020 further expanded our understanding by discovering eight novel susceptibility loci for SCC: rs10399947 (1q21.3), rs10200279 (2q33.1), rs10944479 (6q15), rs7834300 (8q23.3), rs1325118 (9p23), rs7939541 (11p15.4), rs657187, rs11170164 (12q13.13), and rs721199 (12q23.1). These loci involve genes that play roles in cancer progression (SETDB1: rs10399947, CASP8/ALS2CR12: rs10200279, WEE1: rs7939541), immune regulation (BACH2: rs10944479), keratinocyte differentiation (TRPS1: rs7834300, KRT5: rs11170164 and rs657187), and pigmentation (TYRP1: rs1325118). Together, these 22 loci explain approximately 8.5% of the heritable risk associated with SCC [61]. Several histological patterns, including low-risk and high-risk variants, and uncommon variants have been described in Table 3.

The aim of the therapy selected for SCC is to ensure the complete removal and destruction of the primary tumor in order to prevent metastasis. In the latest years, enhancements in tumor staging systems have allowed a more precise stratification of tumors into high- and low-risk classes. There are currently different protocols for managing SCC: curettage and cautery, cryosurgery, radiotherapy, photodynamic therapy, topical imiquimod, laser surgery, conventional surgical excision, and Mohs micrographic surgery [62]. For the majority of SCC, surgical excision with a preset of the clinical margin is the recommended treatment. Recommendations on safety margins in SCC surgical excision vary according to the risk profile of each tumor. For small (<1 cm), slow-growing, well-differentiated SCC on sun-exposed sites, destructive therapies can offer a real alternative to surgical excision [63]. For clinically well-defined, low-risk tumors, less than 2 cm in diameter, a margin of 4 mm will achieve histological clearance in over 95% of cases [64,65]. The high-risk tumors of more than 2 cm in diameter, moderately or poorly differentiated, which extend into the hypodermis or are localized on high-risk sites such as the scalp, ears, lip, eyelid, or nose need margins of 6 mm or greater and a histological examination of the margins [47,65]. Mohs micrographic surgery, with adequate margins, is recommended especially for high-risk squamous cell carcinoma, but it has some disadvantages which include the length of the procedure and the requirement for special equipment and training along with a relatively high cost. For incompletely excised high-risk SCC, the appropriate management includes re-excision to reduce the risk of recurrence and metastasis [65]. Destructive techniques include the following:Curettage and cautery are used sometimes to treat small (<1 cm) low-risk SCC with exceptional cure rates. They are not indicated in recurrent or high-risk tumors.Cryosurgery has disadvantages such as scarring, difficulty in assessing recurrence, and a lack of tissue diagnosis or proof of tumor clearance. For small histologically confirmed SCC, good short-term cure rates have been achieved but a prior biopsy is necessary to establish the diagnosis histologically. It is not applicable for locally recurrent disease or high-risk tumors.Photodynamic therapy is not recommended for SCC [65].

Radiotherapy (RT) is an alternative to surgery for primary SCC if surgery is contraindicated, or for elderly or frail patients, or if RT is likely to lead to a superior cosmetic or functional outcome according to the anatomical site, or if it is the patient’s choice. If the lesion is unresectable and not responsive to RT, then standard systemic treatment choices include chemotherapy or targeted therapy with epidermal growth factor receptor inhibitors. Clinical practice has proven that responses are frequently of a short duration and are sometimes associated with major side effects [66,67].

For advanced and metastatic cSCC, systemic treatment options with a curative intent include immune checkpoint inhibitors, epidermal growth factor receptor (EGFR) inhibitors, and chemotherapy/electrochemotherapy, but this requires a multidisciplinary decision approach. Currently, the sole approved systemic treatment options for certain conditions are inhibitors targeting the programmed death receptor-1 (PD-1). Specifically, cemiplimab and pembrolizumab have been authorized for use in clinical settings [5].

#### 2.2.3. Cutaneous Malignant Adnexal Neoplasms (CMANs)

CMANs are a group of rare and aggressive tumors that originate from the adnexal structures of the skin, including hair follicles, sweat glands, and sebaceous glands [68]. These tumors exhibit malignant behavior and can pose diagnostic challenges due to their diverse histopathological features [69,70].

It is known that certain genetic syndromes, such as Muir–Torre syndrome and Brooke–Spiegler syndrome, are associated with an increased risk of developing CMANs [71]. These syndromes are caused by germline mutations in genes such as MLH1, MSH2, and CYLD, and are also linked to other malignancies such as colorectal cancer and breast cancer [72]. The Wnt/-catenin, Hedgehog, nuclear factor B, and Hippo intracellular signaling pathways are among the intracellular signaling pathways that are modulated by oncogenic drivers of adnexal neoplasms and that can serve as therapeutic targets [73]. Prolonged and excessive exposure to ultraviolet (UV) radiation is considered a risk factor for the development of some CMANs, particularly sebaceous carcinoma. UV radiation can lead to DNA damage and genetic alterations, contributing to the initiation and progression of these tumors. Individuals with compromised immune systems, such as organ transplant recipients or those with HIV/AIDS, have an increased risk of developing CMANs. Immune suppression impairs the body’s ability to control the growth of abnormal cells, facilitating the development of malignancies [74].

The accurate diagnosis of patients necessitates a specialized pathological examination. Sebaceous carcinoma, for instance, shows diverse histological patterns, including lobular, papillary, and poorly differentiated variants. They often exhibit infiltrative growth, marked cytological atypia, and sebaceous differentiation with characteristic vacuolated cells [75].

Malignant sweat gland tumors encompass sweat gland carcinomas, such as eccrine carcinoma and apocrine carcinoma. Histologically, these tumors can display various growth patterns, including solid, cystic, and tubular arrangements. Eccrine carcinomas often exhibit uniform cells with an abundant cytoplasm, while apocrine carcinomas may exhibit more pleomorphic cells and decapitation secretion [76].

Tricholemmal carcinomas arise from the outer root sheath of the hair follicle and exhibit atypical cells with a clear cytoplasm and prominent nucleoli. They often demonstrate infiltrative growth and can resemble squamous cell carcinoma, posing diagnostic challenges [77].

Additional rare CMANs include malignant pilomatricoma, malignant cylindroma, and malignant mixed tumors. These tumors exhibit malignant features such as cellular atypia, mitotic activity, and infiltrative growth [69].


*Management*


Wide local excision with clear margins is the primary treatment for CMANs. Adequate resection aims to achieve the complete removal of the tumor while minimizing the risk of local recurrence. In cases where clear margins cannot be obtained due to the tumor’s anatomical location, Mohs micrographic surgery may be employed to ensure complete tumor removal while sparing healthy tissue [78].

In cases where CMANs have a high risk of regional lymph node involvement, such as in aggressive sweat gland carcinomas, sentinel lymph node biopsy can be performed to determine the extent of lymphatic spread. This information guides subsequent management decisions, including lymph node dissection or adjuvant radiation therapy [79].

Adjuvant radiation therapy may be employed in cases with high-risk features, such as incomplete excision, positive margins, or lymph node involvement. Radiation can help to reduce the risk of local recurrence and improve disease control [80].

In advanced or metastatic CMANs, systemic therapies including chemotherapy, targeted therapy, or immunotherapy may be considered. The choice of treatment depends on the specific histopathological subtype, molecular profile, and the presence of actionable targets [81].


*Novelties*


Advances in molecular techniques have shed light on the genetic alterations underlying CMANs. Molecular profiling can provide valuable information regarding tumor biology, prognosis, and potential therapeutic targets. For example, HER2 amplification has been identified in a subset of sebaceous carcinomas, allowing for the exploration of targeted therapies such as HER2 inhibitors [80].

Recent developments in immunotherapy have shown promise in the treatment of advanced CMANs. Immune checkpoint inhibitors, such as pembrolizumab and nivolumab, have demonstrated efficacy in some cases, particularly those with microsatellite instability or programmed death-ligand 1 (PD-L1) expression [82].

Specific genetic alterations, such as mutations in the MAPK pathway, have been identified in CMANs. Targeted therapies, including BRAF and MEK inhibitors, are being investigated as potential treatment options for tumors harboring these mutations [83].

The management of CMANs should be individualized, considering factors such as tumor stage, histopathological subtype, molecular characteristics, and the patient’s overall health. Regular surveillance and follow-up are essential to monitor for local recurrence, lymph node involvement, or distant metastasis. Close collaboration between healthcare providers and ongoing research efforts are key to improving the diagnosis and treatment outcomes for patients with CMANs.

##### Sebaceous Carcinoma (SeC)

SeC is an uncommon and potentially aggressive cutaneous malignancy showing sebaceous differentiation. It accounts for 0.7% of skin cancers and 3–6.7% of cancer-related deaths [75,84]. The tumors typically affect middle-aged or elderly adults, with the median age at diagnosis being 73 years. The incidence has been increasing in recent years, being much more predominant in elderly, White males. This can be, in part, attributed to an improved diagnostic ability, but additional studies are required to establish the cause [85]. Variations of notable significance in incidence rates have been noted across different geographic regions. For instance, the incidence of this condition is approximately 0.41 per million in the UK and 0.65 per 100,000 in Canada. In contrast, in China, it accounts for nearly one-third of malignant eyelid tumors and ranks second in frequency, following basal cell carcinoma. One study in Japan showed that the rate of SeC equalizes that of basal cell carcinoma [86]. SeC can arise in any site, being commonly classified as periocular or extraocular [87]. Extraocular locations are rare, arising mostly in the head and neck, major salivary glands, breasts, oral mucosa, lungs, and ovaries. They may develop in sebaceous naevi [88]. The clinical appearance may be nonspecific. A biopsy is essential to establish a diagnosis and to make a differential diagnosis with benign sebaceous neoplasms, other adnexal tumors, and BCC. In patients with a sebaceous neoplasm, we should also consider a diagnosis of Muir–Torre syndrome [87,88]. The gross appearance of SeC shows a tan-pink or yellowish nodule, sometimes ulcerated, being several centimeters in dimension. Periocular lesions require clinical differentiation from chalazions or chronic blepharoconjunctivitis. Occasionally, they may affect immunocompromised patients. SeC of the eyelid may arise after radiation therapy for retinoblastoma, where 1/3 cases recur, and 25% die of metastases [85,86].

The histopathological patterns vary from well-differentiated to anaplastic tumors. SeC typically shows lobules of variably atypical polygonal cells, with comedo-necrosis, extending into the deep reticular dermis and subcutis. The tumoral stroma is fibrovascular, having inflammatory cells with foreign-body-type giant cells and sebaceous material [84]. According to the proportion of typical sebocytes, tumors may be classified into well, moderately, or poorly differentiated SeC. The well-differentiated neoplasms show cells with an abundant foamy, finely vacuolated cytoplasm and distinct cell borders, vesicular nuclei with distinct nucleoli, and a variable number of mitotic figures. Poorly differentiated SeC show cells with a scant cytoplasm, having indistinct vacuoles, high N:C ratios, nuclear pleomorphism, prominent nucleoli, atypical mitosis, central necrosis, and pagetoid involvement of the covering skin. Tumoral cells are immunoreactive for cytokeratins and EMA [86].

Alterations in the expression of DNA mismatch repair enzymes including MLH1, MSH2, MSH6, and PMS2 may be observed [89] and seem to have a possible relationship to Muir–Torre syndrome [88]. The molecular pathogenesis of SeC is still being discussed. The dysregulation of various signaling pathways, including p53, Wnt/β-catenin, COX, and Hedgehog pathways, has been associated with abnormalities in sebaceous cell carcinoma (SeC). Mutations in genes such as p53 and LEF-1 have also been implicated [90,91,92]. Additionally, specific mutations in CDKN2A, EGFR, CTNNB1, and KRAS have been Identified in certain tumors [93]. Epidemiological evidence indicates that both ocular and extraocular SeC carry a substantial risk, including a 30–40% chance of local tumor recurrence, a 20–25% risk of distant metastasis, and a 10–30% risk of tumor-related mortality. Prognostic indicators of poorer outcomes include orbital or vascular invasion, the involvement of both eyelids, poorly differentiated or multicentric tumors, a large tumor size, infiltrative growth pattern, and pagetoid spread. Approximately 3/4 of all patients with SeC survive for more than 5 years after diagnosis. Prognosis is dependent of the size of the primary tumor, with the margin removed having a higher risk of recurrence indicated by a 1–3 mm margin versus 5 mm. The removal of the tumor may also be enhanced through staged excision using margin control [94].

Early intervention is crucial in the management of sebaceous cell carcinoma (SeC) due to its propensity to spread to regional lymph nodes. The primary approach for treating sebaceous carcinoma continues to be surgical intervention, with options such as Mohs micrographic surgery offering the advantages of comprehensive margin assessment and reduced recurrence rates. In more advanced cases, radiation therapy, chemotherapy, or a combination of therapies may be employed to achieve optimal treatment outcomes [87].

#### 2.2.4. Primary Cutaneous Lymphomas (PCLs)

PCLs are a heterogeneous group of lymphoproliferative diseases, with no evidence of extracutaneous determination at the moment of the diagnosis. PCLs usually have a very different evolution, prognosis, and treatment compared with the lymphomas that may secondarily involve the skin. The classification of PCLs separates them into two important groups: cutaneous T-cell lymphomas (approximately 75% of all PCL) and cutaneous B-cell lymphomas (approximately 25% of PCL in the Western World). In 2018, the World Health Organization–European Organization for Research and Treatment of Cancer (WHO–EORTC) revised and updated the classification of primary cutaneous lymphomas, adding new types of primary cutaneous lymphomas (Table 4) [6].

According to estimates from the National Cancer Institute’s Surveillance, Epidemiology, and End Results (SEER) registry, cutaneous B-cell lymphomas (CBCL) are the least common among primary cutaneous lymphomas (PCL), with an incidence of 3.1 per million from 2001 to 2005. However, they account for 29% of all PCL cases. The annual incidence rate of CBCL has steadily increased, reaching 3.92 per million between 2006 and 2010. The age-adjusted incidence of all types of cutaneous T-cell lymphomas (CTCL), based on different SEER datasets, ranged from 6.4 to 7.7 million. Both CBCL and CTCL have consistently and dramatically increased in incidence over the past three decades, with CTCL increasing by 2.9 per million per decade. Based on available data, it is estimated that over 3000 new patients are diagnosed with PCL each year [6].

##### The Cutaneous T-Cell Lymphomas (CTCL)

Mycosis fungoides (MF) is the predominant type of cutaneous T-cell lymphoma (CTCL), accounting for 60% of all CTCL cases. Although Sézary syndrome (SS) is classified as a distinct entity from MF, it shares similar histologic criteria and staging with MF and often arises from MF. The prevalence of MF is estimated to be over 50,000 based on survival data. Patients with MF who have tumor involvement or nodal involvement experience compromised 10-year survival rates of 42% and 20%, respectively. In one study, patients with SS, characterized by blood involvement and potential nodal involvement, had a 5-year survival rate of 24% [95].

In the last classification of the WHO–EORTC, published in *Blood* in 2019, they recognized new and distinct types of MF: folliculotropic MF (FMF), pagetoid reticulosis, and granulomatous slack skin. The recent studies reveal that there is a small group of FMF with an excellent prognosis [6].

SS, this rare leukemic type of CTCL, clinically defined by erythroderma, pruritus, generalized lymphadenopathy, and histologically by the presence of Sézary cells in the skin, lymph nodes, and peripheral blood, is very difficult to differentiate from other erythrodermic inflammatory dermatoses (EIDs) [96,97,98,99,100]. For a positive diagnosis, it is mandatory to use the Sézary cells as evidence (more than 1000 μL) or an expanded CD4+T-cell population, with CD4/CD8 ratio ≥ 10, CD41/CD72 cells ≥ 40%, or CD41/CD262 cells ≥ 30%. PD-1 (CD279) and KIRDL2 (CD158k) are the newest biomarkers described by the studies, and they can be used for the differential diagnosis between SS and EIDs, both in skin and in the peripheral blood [6]. Primary cutaneous CD30+T-cell LPD account for approximately 25% of CTCL, represented by lymphomatoid papulosis (LyP) and primary cutaneous anaplastic large lymphoma (C-ALCL). The new WHO–EORTC classification introduces two more recently described types of LyP: type D (mimicking primary cutaneous aggressive epidermotropic cytotoxic T-cell lymphoma [96]) and type E (angiodestructive, angiocentric), with a clinical presentation of large, necrotic, eschar-like lesions [89]. A genetic characteristic of this is the presence of chromosomal rearrangements in the DUSP-IRF4 locus on 6p25.3 [98].

CTCL other than MF, SS, PC CD30+ LPD represent less than 10% of CTCL, and newly introduced in the classification of WHO–EORTC are chronic active Epstein–Barr virus (EBV) infections in childhood, primary cutaneous acral CD81 T-cell lymphoma, and primary cutaneous CD41 small/medium T-cell LPD [6]. EBV plus LPD in childhood includes hydroa vacciniforme-like LPD (HV-like LPD), with a CD81 T-cell phenotype and hypersensitivity reactions to mosquito bites, with a NK-cell phenotype, both with a risk of progression to systemic EBV1T- or natural killer (NK)-cell lymphoma [101]. Primary cutaneous acral CD8+ T-cell lymphoma is a newly described type of lymphoma, histologically characterized by a diffuse infiltrate of medium-sized CD8^+^ cytotoxic T cells, with clinically indolent behavior, in contrast with histologically aggressive malignant aspects [102]. The atypical cells show a CD31, CD42, CD81, and CD302 T-cell phenotype with variable losses of pan-T-cell antigens (CD2, CD5, CD7), TIA-1 positive, and cytotoxic proteins (granzyme B, perforin), which are negative [103]. CD68 often shows a positive Golgi dot-like staining [104]. EBV is negative. Primary cutaneous CD41 small/medium T-cell LPD was described first in 2005, as a provisional type of CTCL, histologically characterized by a predominance of small- to medium-sized CD41 pleomorphic T cells, and clinically by a solitary plaque or tumor, localized on the face, neck, or upper trunk, which has an excellent prognosis, with no staging necessary. The proliferation rate is low, varying between 5% and at most 20% [6].

##### Cutaneous B Cell Lymphomas (CBCL)

The WHO–EORTC classification from 2005 proposes three groups of CBCLs: primary cutaneous marginal zone lymphoma (PCMZL), primary cutaneous follicle center lymphoma (PCFCL), and primary cutaneous large B-cell lymphoma, leg type (PCDLBCL, LT) (Table 5) [6]. PCMZLs, part of the MALT lymphomas group, with an excellent prognosis of an approximately 100% 5-year disease-specific survival rate, present usually with cutaneous involvement, with extracutaneous dissemination being rarely observed. PCMZL demonstrates the expression of class-switched immunoglobulins (IgG, IgA, and IgE), while it lacks the expression of the chemokine receptor CXCR3, which plays a crucial role in guiding the migration of malignant B cells [6].

The optimal management of patients with cutaneous B-cell lymphomas (CBCL) requires a multidisciplinary approach involving dermatology, medical oncology, and radiation oncology. For patients with a solitary lesion, radiation therapy is recommended, while patients with multiple lesions may consider radiation therapy or deferring radiation as viable options. Studies have shown a high rate of local control (98%) for indolent CBCL treated with radiation alone. Conversely, surgical excision alone has been associated with a 25% local recurrence rate requiring subsequent radiation therapy [105,106]. Therefore, complete excision without radiation, with the option of deferring radiation until disease recurrence, is also considered a reasonable approach. Additional treatment modalities such as topical therapies, intralesional corticosteroids, or rituximab may be considered, particularly for patients with limited skin involvement. In cases of more extensive skin involvement, single-agent rituximab has shown effectiveness in managing the disease [105].

#### 2.2.5. Soft Tissue Sarcomas (STS)

##### Cutaneous Angiosarcoma (CAS)

Primary CAS is a rare (2% of STSs) and aggressive skin cancer with rapid metastasis and poor prognosis, with the mean 5-year survival rate being 33.5% [107]. Secondary angiosarcoma is related to radiation treatment (radiation-associated angiosarcoma) or chronic lymphedema (Stewart–Treves syndrome). The oncogenesis of angiosarcoma may be explained by the mutations of protein tyrosine phosphatase, receptor type B (26% of the patients), and the mutation of phospholipase C, gamma 1 (9%) [108]. Another 36% of these cancers may be explained by *NUP160-SLC43A3* gene fusion [109] and other proportions can be explained by numerous mutations including *KRAS*, *HRAS*, *NRAS*, *BRAF*, *MAPK1*, and *NF1* [110,111]. Recent studies have investigated the role of survivin and forkhead box M1 as potential markers and therapeutic targets for CAS [112,113], and others the role of miR-497-5p or miR-210 downregulation in the pathogenesis of CAS [107]. There is a lack of standardized treatment options; the standard management is surgical resection with wide-margin excision, with or without postoperative radiotherapy or chemotherapy (paclitaxel or doxorubicin as the first line, followed by pazopanib, eribulin mesylate, bevacizumab, or trabectedin) [114,115]. Recently, propranolol treatment or immunotherapy were proposed for CAS management [107]. There are also studies showing that patients with angiosarcoma may benefit from anti-PD-1 (programmed death ligand-1) therapy or anti-tumor mutational burden high (TMB-H) [107,116,117].

##### Kaposi Sarcoma (KS)

KS is a rare type of cancer that primarily affects the skin and mucous membranes. It was first described by Dr. Moritz Kaposi in the late 19th century, and, since then, significant progress has been made in understanding its etiology, histopathology, management, and recent developments in the field [118]. The etiology of Kaposi sarcoma is closely linked to an infectious agent known as Human herpesvirus 8 (HHV-8), also called Kaposi sarcoma-associated herpesvirus (KSHV). HHV-8 is transmitted through various routes, including sexual contact, blood transfusion, and organ transplantation. The virus is more prevalent in certain geographic areas, such as sub-Saharan Africa and the Mediterranean region [119,120].

KS is particularly associated with individuals who have a weakened immune system, such as those with HIV/AIDS. In fact, the epidemic form of KS, known as AIDS-associated KS, was one of the first recognized AIDS-defining malignancies. HHV-8 infection and the resulting dysregulation of the immune system are crucial factors in the development of KS. However, not all HHV-8-infected individuals develop KS, suggesting that additional cofactors, such as genetic susceptibility, may play a role [119].

The clinical manifestation of KS varies greatly from person to person, with some showing latent, indolent illness and others showing tumors that are more aggressive [121].

Histopathologically, KS lesions exhibit distinct characteristics. They typically consist of abnormal blood vessels, inflammatory cells, and spindle-shaped tumor cells. These spindle cells are derived from endothelial cells and are considered the neoplastic cells in KS. The histopathological appearance of KS can vary among different clinical variants, including classic, endemic, epidemic, and iatrogenic forms. In almost all cases, endothelial proliferation is monomorphic with minimal to no atypia. The anaplastic variety of KS should be taken into consideration in the case of significant atypia [121]. Classic KS primarily affects elderly men of Mediterranean or Eastern European descent, while endemic KS occurs in specific regions of Africa. Epidemic KS, as mentioned earlier, is associated with HIV/AIDS, and iatrogenic KS is seen in individuals undergoing immunosuppressive therapy following organ transplantation [122].

The management of Kaposi sarcoma depends on several factors, including the extent of the disease, the individual’s immune status, and the presence of other associated conditions. In the context of HIV/AIDS, antiretroviral therapy (ART) plays a central role. By controlling the underlying HIV infection, ART restores immune function, leading to the regression of KS lesions in many cases [123].

For localized disease, treatment options may include surgical excision, radiation therapy, or intralesional chemotherapy. Surgical excision is often employed for small, solitary lesions, while radiation therapy can be effective for localized or symptomatic lesions [124,125]. Intralesional chemotherapy, such as injections of vinblastine or bleomycin directly into the lesions, can be used to target specific areas.

In more extensive or symptomatic KS, systemic therapies are utilized [126]. Traditional chemotherapy agents, such as liposomal anthracyclines (e.g., liposomal doxorubicin), have shown efficacy in controlling disease progression. However, they may be associated with significant side effects [127]. Targeted therapy has emerged as a novel approach in the management of KS. Tyrosine kinase inhibitors, such as sunitinib and pazopanib, have demonstrated effectiveness by inhibiting angiogenesis and tumor growth.

Immunotherapy has also shown promise in the treatment of advanced or refractory KS [128]. Immune checkpoint inhibitors, such as pembrolizumab, have demonstrated encouraging results by unleashing the immune system to recognize and attack cancer cells. This approach holds great potential for improving outcomes in KS patients [129].

Furthermore, ongoing research is focusing on developing HHV-8-specific therapies. These antiviral agents aim to specifically inhibit viral replication and disrupt HHV-8-associated signaling pathways, potentially impacting KS progression. Several novel compounds are being investigated in preclinical and early clinical trials, offering hope for future therapeutic options [130].

##### Dermatofibrosarcoma Protuberans (DFSP)

DFSP is a rare type of soft tissue sarcoma that primarily affects the skin. It is characterized by the abnormal growth of fibroblast cells in the dermis, leading to the formation of a protuberant tumor. Regarding the etiopathogeneses, it is known that the majority of DFSP cases are associated with a specific chromosomal translocation, t(17;22) (q22;q13), which leads to the fusion of the COL1A1 and PDGFB genes. This fusion gene results in the overexpression of platelet-derived growth factor beta (PDGFβ), a potent mitogen that drives the proliferation of fibroblast cells [131].

In rare cases, DFSP can be associated with familial predisposition, suggesting the involvement of additional genetic factors. The morphology of DFSP typically presents as a slow-growing, indurated (firm) plaque or nodule on the skin, most commonly on the trunk or extremities. Microscopically, DFSP is characterized by a storiform pattern, with spindle-shaped fibroblast-like cells arranged in a swirling pattern. These cells infiltrate the surrounding dermis and subcutaneous tissue. Additionally, the presence of a characteristic “honeycomb” pattern, representing fibroblastic islands within the tumor, is a diagnostic feature [121].

Immunohistochemical staining for CD34 is commonly used to confirm the diagnosis of DFSP, as most cases show positive staining for this marker.

The primary treatment for DFSP is surgical resection with clear margins. The goal is to completely remove the tumor while preserving normal tissue. This approach offers the best chance for long-term disease control. Given the locally infiltrative nature of DFSP, Mohs surgery is often employed, especially in cases where the tumor is located in cosmetically sensitive or functionally important areas [121,132]. Mohs surgery allows for the precise removal of the tumor while sparing healthy tissue, as the surgeon can examine the entire surgical margin microscopically during the procedure [121].

In cases where the tumor has infiltrated deep into the underlying tissue, or if there is a high risk of recurrence, adjuvant therapies such as radiation therapy may be recommended to further reduce the risk of local recurrence. In advanced or metastatic DFSP, systemic therapies, including tyrosine kinase inhibitors (such as imatinib), may be considered. These targeted therapies aim to inhibit the PDGFβ receptor and have shown promising results in controlling disease progression [131].

Recent advancements in understanding the molecular basis of DFSP have paved the way for the development of novel targeted therapies. For instance, the use of imatinib, a tyrosine kinase inhibitor, has shown promising results in treating locally advanced or metastatic DFSP, particularly in cases with PDGFβ receptor activation [133].

Minimally invasive surgical techniques, such as robotic-assisted surgery and laparoscopic surgery, are being explored for the management of DFSP. These approaches aim to achieve complete tumor removal with improved cosmetic outcomes and reduced morbidity [134].

In addition, the identification of specific genetic alterations, such as the COL1A1-PDGFB fusion gene, has allowed for a better understanding of the molecular mechanisms driving DFSP. This knowledge opens the door for personalized medicine approaches, where targeted therapies can be tailored to the specific genetic profile of each patient’s tumor [135].

It is important to note that the management of DFSP should be individualized, and treatment decisions should be made in consultation with healthcare professionals experienced in managing the disease. Regular follow-up and monitoring are crucial for detecting any signs of recurrence or metastasis.

##### Cutaneous Leiomyosarcoma (LMS)

LMS is a rare malignant neoplasm of muscular origin, accounting for 2–3% of all cutaneous soft tissue sarcomas and 0.04% of all neoplasms. The incidence rate is 0.2/100,000/year. It is the third most common cutaneous sarcoma after dermatofibrosarcoma protuberans and Kaposi sarcoma [136]. The ratio of men to women is 3:1. Cutaneous SML occurs predominantly in the 50–70 years age group (mean 62 years) [137] compared to subcutaneous SML which occurs in the 50–80 years group [127]. It most commonly affects the White population (92%) [138].

The etiology remains unknown. Among the most common predisposing factors are leiomyomas (as precancerous lesions), and a history of local injury/trauma [136]. At the molecular level, the overexpression of tyrosine kinase receptors (IGFR and PDRF) has been observed. The majority of SMLs originate from muscle cells in the smooth muscle pillar. In exceptional cases, it can develop from vascular structures [137]. The spectrum of clinical manifestations is non-specific, and the most common manifestations are a single, firm, pink, smooth-surfaced lump, or a red-brown exophytic tumor [136]. Rare manifestations include a plaque formed by multiple indurated nodules, and grouped asymptomatic lesions of the lower limbs [137].

LMS can be classified into three clinico-pathological groups: the purely cutaneous (dermal) form, the subcutaneous (hypodermal) form, and cutaneous metastases of extracutaneous leiomyosarcoma [136]. Cutaneous LMS has a slow growth; its initial size is between 1 and 3.5 cm, it is painful on palpation (63%), and can sometimes be associated with itching, burning, and paresthesia in the lesion territory. Subcutaneous CML is well demarcated, lipoma-like, solid in consistency, and large in size. Regarding cutaneous LMS localization, 50% of cases are located on the extensor surfaces of the lower limbs (regions with an increased density of hair follicles and arrector pili muscles); less frequently, they may appear on the scalp, facies, trunk, lips, genital area (scrotum, vulva, penis), and gluteal area [135].

They are histologically characterized by the dermal proliferation of elongated spindle cells, arranged in interpenetrating bundles, with cigar-shaped nuclei and an eosinophilic cytoplasm. There are two architectural patterns: the nodular pattern (characterized by high cellularity, atypia, mitosis) and the diffuse pattern (characterized by low cellularity, lower mitotic load, cytological atypia, infiltrative growth, and mitotic activity (>1 mitosis/10 high power fields)) [137].

A positive diagnosis is made on the basis of a skin biopsy, which must include subcutaneous cellular tissue; immunohistochemical studies (positive for vimentin, desmin, h-caldesmon, muscle-specific actin, α-actin of smooth muscle, and smooth muscle myosin); and sometimes on the basis of S-100 protein and cytokeratins. Undifferentiated lesions and subcutaneous LMS are negative for desmin. The diffuse overexpression of p53 present in cutaneous LMS vs. absent in leiomyoma helps to make the differential diagnosis. At least two muscle immunohistochemical markers are required for diagnosis [136].

The preoperative imaging approaches differ depending on the specific subtypes of LMS. For cutaneous SML, a preoperative MRI is performed for large, infiltrative lesions or those in hard-to-reach places like the head. A chest X-ray is also useful. For Subcutaneous SML, MRI and thoracoabdominal CT are always performed to differentiate metastatic cutaneous SML from deep tissue SML. A thoracoabdominal CT is a suitable staging test for patients suspected to have disseminated disease [137].

Regarding the treatment, complete surgical excision is the preferred treatment option [139]. However, there is no consensus on the appropriate safety limits. Traditionally, aggressive surgery with 3–5 cm margins has been recommended [136]. Recently, conservative surgery with 1 cm margins has shown similar results, with no increase in the local recurrence rate and improved morbidity. The depth of excision includes the fascia, and, in infiltrated cases, the muscle. Up to 20% of patients may require repeat excisions to achieve negative margins. Mohs surgery is a potential future approach [138]. The role of adjuvant radiotherapy is uncertain. It is useful for deep/large lesions (>5 cm), when there are unfavorable prognostic factors present, and for positive surgical margins. Radiotherapy (with chemotherapy) is used for unresectable cases but has mixed success rates. Radiotherapy remains the best solution for local palliative control in tumors with metastases. Neoadjuvant chemotherapy is used for disseminated disease and includes various drugs such as doxorubicin, ifosfamide, gemcitabine, docetaxel, taxotere, dacarbazine, and trabectedin [139]. Future directions in treatment include immunotherapy, gene therapy, and targeted therapies with tyrosine kinase inhibitors.

Distant metastases often occur in the lungs through hematogenous dissemination. Superficial LMS can metastasize to the scalp, while regional lymph node involvement is rare [137]. Post-surgical monitoring involves clinical examinations every 4 months during the first 2 years to detect local recurrences, then every 6 months until the 5th year after surgery, and annually thereafter until 20 years for detecting late recurrences. Chest X-rays are performed annually during the first 5 years after surgery, along with general clinical and locoregional lymph node assessments. MRI is occasionally used for recurrent or hypodermic lesions [139].

The most important prognostic factors for cutaneous SML are size, location, and the degree of histological differentiation [136]. Tumor size is an independent prognostic factor associated with decreased survival. Tumor size (≥5 cm), a deep location with fascial involvement, and a high histological grade are correlated with decreased survival. A study by Svarvar et al. (2007) identified tumor depth as a significant prognostic factor for metastasis, local recurrence, and death [140]. A recent study (2021) found that the presence of metastases is the strongest risk factor correlated with death. The 5-year survival rate is over 95% for cutaneous SML and 65% for subcutaneous SML. Cutaneous SMLs carry a risk of recurrence in the absence of complete excision, but they have low metastatic potential [136].

#### 2.2.6. Merkel Cell Carcinoma (MCC)

MCC is a rare neuroendocrine tumor of the skin, described as a fast growing, painless, reddish-purple nodule, with a rapidly progressive course, aggressive behavior, and is seen mainly in the elderly [141]. MCC is commonly localized on the facial skin, on extremities, usually on UV-exposed areas, in the Caucasian elderly population, and is associated with intense sun exposure and with Merkell cell polyomavirus (MCPyV) [141,142,143]. The incidence of MCC is low, but in the US there has been an increase in cases (95% from 2000 to 2013), and it is also increasing in other Western countries and in Australia [141].

The clinical negative prognostic factors are tumor burden (>1 cm), regional metastases, gender (male), localization on the upper part of the body (head and neck primary site) [133], but also chronic T-cell immunosuppression, HIV, chronic lymphocytic leukemia (CLL), solid organ transplant, and the presence of lymphovascular invasion (LVI) [142,143].

The acquisition of histopathological identification, with specific immunohistochemical markers, has contributed to a better diagnosis, but still there are some histopathological negative prognostic factors such as depths of invasion, T-cell infiltration, and lymph vascular invasion [141]. The association with MCPyV is very important, and is seen in more than 80% of cases from Western countries [141], being one of the indicators for recurrence [134]. The histopathological confirmation of MCC requires immunohistochemistry, and molecular markers such as neuroendocrine markers (chromogranin A, synaptophysin, CD56), cytokeratin 20 (dot-like pattern), neuron-specific enolase (NSE), and MCPyV large T cell antigen (LT) (CM2B4) [141,142,143]. For differential diagnosis (with melanoma, small cell lung cancer, lymphoma), S-100, thyroid transcription factor-1 (TTF1), and leukocyte common antigen (LCA) can be used, respectively [141].

The MCPyV seroprevalence is very high in the general population, increasing with age (50% in childhood versus 80% in adults over 50), and with latent infection in immunocompetent hosts [141,142,143]. The implication of MCPyV in the oncogenesis of MCC is connected to two factors: the clonal integration of the viral genome in the host genome, and a mutation that implicates the loss of expression of the C-terminus of the “large T antigen”, which must inhibit the expression and synthesis of the oncoproteins’ large T antigen and small T antigen that promote cell cycle progression and survival.

Diagnosis is made on the basis of clinical examination. The current NCCN practice guideline from 2023 does not recommend routine baseline imaging, but, in the case of primary advanced tumors, an MR-scan or CT-scan can be considered for the evaluation of deeper tissue invasion and is encouraged for the staging of most cases of MCC. Even though there are some studies that indicate that whole-body PET with fused axial imaging is more sensitive, the NCCN guide recommends CT with a contrast of chest/abdomen/pelvis (and neck/brain if there is clinical suspicion) as an acceptable alternative [142].

According to the NCCN guidelines from 2023, the recommended treatment approach for MCC includes node dissection and radiotherapy (RT) [142,143]. Sentinel lymph node biopsy (SLNB) is an important tool for staging and treatment. The surgical excision margin should be determined based on the risks and potential complications, with the primary objective of achieving tissue closure. Adjuvant RT should be administered promptly. Adjuvant chemotherapy is not recommended for primary MCC [142,143]. In cases of recurrent locally advanced disease where curative surgery and RT are not feasible, pembrolizumab can be used. Retifanlimab-dlwr can also be considered if the patient is not suitable for surgery or RT. For disseminated disease (M1), avelumab, nivolumab, and pembrolizumab can be used. Retifanlimab-dlwr is another recommended regimen if the patient cannot undergo surgery or RT [142,143].

## 3. Conclusions

Comprehensive treatment is necessary for skin cancer. New information regarding the treatment of these disorders is covered in the updated guidelines. The use of dermoscopy by experienced dermatologists can improve the accuracy of diagnosis. In addition, several new technologies are aimed at improving the accuracy of pre-biopsy diagnosis: image analysis using artificial intelligence, three-dimensional imaging of the whole body, reflective confocal microscopy, optical coherence tomography, and the retrieval of epidermal genetic information from adhesive tape stripping. New studies have revealed that specific gene expression patterns in melanoma cells can provide valuable insights into the likelihood of metastasis in stage I or II melanomas. Building upon this research, a laboratory test called DecisionDx-Melanoma has been developed and is currently accessible for clinical use. There are new insights in the staging of melanoma, and also in the management recommendations, for the adjuvant and systemic treatment of the disease.

In addition, dermoscopy contributes to establishing the diagnosis of BCC and SCC with an accuracy of 95–99% to differentiate BCC and SCC from other neoplastic and inflammatory disorders, and it helps to differentiate the subtypes before skin biopsy. Reflectance confocal microscopy (RCM) and optical coherence tomography (OCT) are novel noninvasive diagnostic techniques for BCC and SCC diagnosis. Several tumor suppressor genes and proto-oncogenes have been described in BCC pathogenesis, including the components of the Hedgehog pathway, Patched Homolog 1 (PTCH1), smoothened (SMO), the tumor protein p53 (TP53), and members of the RAS proto-oncogene family. In the pathophysiology of SCC, molecular pathways such as RAS/RAF/MEK/ERK and PI3K/AKT/mTOR play a significant role. Surgery remains the gold standard for both BCC and SCC, but, in recent years, for advanced and metastatic cSCC, systemic treatment options with a curative intent include immune checkpoint inhibitors, epidermal growth factor receptor (EGFR) inhibitors, and chemotherapy/electrochemotherapy; however, this requires a multidisciplinary decision approach.

Surgical treatment, particularly Mohs micrographic surgery, continues to be the primary approach for managing sebaceous carcinoma. This method offers advantages such as comprehensive margin evaluation and low rates of recurrence. In cases that have progressed to an advanced stage, alternative options such as radiation therapy, chemotherapy, or a combination of therapies may be employed.

A diverse group of lymphoproliferative disorders, primary cutaneous lymphomas (PCLs), have no extracutaneous determination at the time of diagnosis. In contrast to lymphomas that may secondarily involve the skin, PCLs often have a significantly distinct progression, prognosis, and treatment. In 2018, the World Health Organization–European Organization for Research and Treatment of Cancer (WHO–EORTC) updated the classification of primary cutaneous lymphomas, adding new types of primary cutaneous lymphomas. Effective patient management necessitates a comprehensive approach involving multiple disciplines, including dermatology, medical oncology, and radiation oncology.

## Figures and Tables

**Table 1 ijms-24-11176-t001:** Management of Local/Locoregional Disease.

Local Disease		
Wide local excision (WLE) (II,B) [1]:	0.5 cm—in situ melanoma; 1 cm—Breslow < 2 mm; 2 cm—Breslow ≥ 2 mm.	
Acral/facial melanoma	Safety margins are decreased to ensure preservation of function/slow Mohs technique.	
Lentigo maligna melanoma	Radiotherapy	An option to avoid extensive surgery
SNB (sentinel node biopsy)	Is recommended for precise diagnosis and staging in melanoma stage pT1b or higher (tumor thickness > 0.8 mm/tumor thickness < 0.8 mm with ulceration (IIB); is not recommended for pT1a melanomas [16]	Can add to the accuracy of the prognosis [1].
**In-transit disease** [1]		
Surgery	Resectable satellite or ITM patients	Isolated case
Adjuvant radiotherapy (RT) can be Considered for the following:	Cases of inadequate resection of LMM (lentigo malignant melanoma) R1 resections (microscopic tumor at the margin) of MM metastases (only when second surgery is not adequate) After resection of bulky disease	
**Systemic therapy**		
Adjuvant systemic therapy	IFN-α—is no longer routinely proposed in the adjuvant setting. Ipilimumab—monoclonal antibody blocking cytotoxic T lymphocyte-associated antigen 4 (CTLA-4). Dabrafenib/trametinib—anti-programmed cell death protein 1 (anti-PD-1) (I,A). Nivolumab (anti-PD-1)—shows benefits for stage III B/C, IV (AJCCC seventh edition) resected melanoma. It shows fewer grade 3/4 adverse events compared with toxic high dose ipilimumab [17]. Pembrolizumab (I,A) [18].	Recommended for high-risk primary melanoma (stage II B/C) and completely resected lymph node metastases (stage III).
Targeted therapy–adjuvant targeted therapy	Dabrafenib/trametinib (one of the established treatment options for adjuvant BRAF-mutated melanoma)	Was approved by EMA for adjuvant treatment for melanoma in August 2018 (I,A; ESMO-MCBv1.1 score:A);
**Advanced/metastatic disease**		
Surgical or ablative treatment of resectable stage IV	Is an option only for selected patients, preferably combined with adjuvant systemic therapy [19].	
Systemic treatment of unresectable stage III and IV disease with immunotherapies or targeted therapies:	PD-1 blockade: nivolumab/pembrolizumab PD-1 blockade (nivolumab) combined with CTLA-4 blockade (ipilimumab) BRAF V600-mutated melanoma–(II,B): BRAF inhibition (vemurafenib/dabrafenib/encorafenib) + MEK inhibition (cotimetinib/trametinib/binimetinib) [20]. stage IIIB/C, IVM1a—talimogene laherparepvec (T-VEC) (I,B) [1].	Current treatment decision is based on several parameters but they need to be individualized for each patient [21].

**Table 2 ijms-24-11176-t002:** The clinico-anatomical features and risk of recurrence of basal cell carcinoma subtypes.

Histological Subtypes	Anatomical Area	Clinical Features and Risk of Recurrence	Histopathological Features
Superficial (10–30%) [22,30,31]	Trunk	Erythematous, mildly scaly, well-circumscribed plaque/patch, lower risk of recurrence	Numerous nests of tumor cells having a basaloid palisading feature in periphery, connected to the epidermis; myxoid stroma; inflammatory lichenoid band-like infiltrate
Nodular (50–80%) [22,32]	Head and neck	Pearly plague or nodule with a smooth surface-rolled borders and telangiectasis, with lower risk of recurrence (rare metastasis)	Malignant basaloid cells forming small groups in the dermis, with peripheral palisading; mucoid stroma containing plump spindle cells
Fibroepithelial [22,33]	Trunk	Flesh-colored nodule, lower recurrence	A reticular pattern of epidermal basaloid keratinocytes disposed in thin strands; spindle cell stroma
Sclerosing/morphoeic [22,34]	Head and neck	Poorly-defined, scar-like plaque borders that rarely ulcerate or bleed; higher local recurrence and perineural invasion	Basaloid cells disposed in thin cords without peripheral palisading bound by a sclerotic collagenous stroma; tumor stroma is positive for smooth muscle alpha-actin
Micronodular (6–64%) [22]	Head and neck	Elevated/flat, poorly define lesion; high risk of recurrence	Small groups of basaloid cells within the dermis, with delicate peripheral palisading associated with retraction artefact
Infundibulocystic [31]	Head and neck	Well-circumscribed pearly papules; lower recurrence	anastomosing strands of basaloid cells along with disseminated infundibulum-like cystic structures
Infiltrative [22]	Upper trunk, head, and neck	White/yellow/pale pink plaque which is indurated, flat/depressed and shows crusts/erosions; high-recurrence	Delicate cords of few basaloid keratinocytes with angulated ends; mucinous/myxoid stroma
Basosquamous (6–44%) [35]	Head and neck	papule or nodule which may ulcerate; high-recurrence metastases in lymph node, lung	Nodular or superficial BCC component covering an invasive front which displays BCC and SCC histologic characteristics.

BCC, basal cell carcinoma; SCC, squamous cell carcinoma.

**Table 3 ijms-24-11176-t003:** Histological patterns, including low-risk and high-risk variants, and uncommon variants of SCC.

Histological Subtypes	Anatomical Area	Clinical Features	Histopathological Features
Keratoacanthoma	Facial skin, the dorsum of the hands and forearms (men) and on the legs (women) in elder, fair-skinned individuals	Symmetrical, regular dome-shaped lesion, covered with keratin	*Early proliferative stage*—symmetrical lesion composed of invaginations of squamous epithelium. *Mature stage*—typical crateriform architecture with well-differentiated squamous lobules displaying follicular differentiation with trichilemmal keratinization surround a central keratin core. *Regressing stage*—a crateriform appearance, but the tumor turns into a thinner, flattened lesion, with less squamous lobules and horn cysts.
Acantholytic squamous cell carcinoma	Head, neck, face, and ears	No typical features	Thickened/hyperkeratotic/ulcerated epidermis; acanthosis of tumor cells; pseudo-glandular pattern; pseudo-vascularization.
Clear cell squamous cell carcinoma	Oral mucosa skin	Hyperkeratotic plaques (palmoplantar); condylomatoid excrescences/nodular (anogenital)	Exophytic and endophytic architecture; prominent hyperkeratosis; deep tongues of intradermal growth; minimal cytological atypia.
Spindle cell squamous cell carcinoma	Face, head, neck, chest, and upper extremities (sun exposed areas); mucocutaneous areas of the head and neck, urogenital tract, distal penis	Similar to conventional SCC	Severe solar elastosis/actinic keratosis/SCC in situ; fascicles of pleomorphic spindle cells, with numerous mitotic activity, closely packed; no keratinization.
Verrucous squamous cell carcinoma	Head and neck	Similar to conventional SCC	Verruciform surface; blunt endophytic growth; malignant cells with cytoplasmic unilocular vacuolation and focal keratinization;
Squamous cell carcinoma with sarcomatoid differentiation	Sun-exposed skin of the face and head, neck, chest, and upper extremities	No typical features	a classic SCC component associated with a sarcomatous component; severe solar elastosis—typical; chondroblastic, osteoblastic, rhabdomyosarcomatous, myoid differentiation.
Lymphoepithelioma-like carcinoma	Sun-exposed skin, most often of the head and neck	No typical features	Poorly differentiated tumor cells surrounded and infiltrated by lymphocytes and plasma cells; resemble undifferentiated nasopharyngeal carcinoma (lymphoepithelioma); requires immunohistochemistry.
Pseudo-vascular squamous cell carcinoma	Head and scalp	No typical features	An infiltrative high-grade SCC made by cords of cells with intercellular lumen-like spaces; stroma may show myxoid change;
Squamous cell carcinoma with osteoclast-like giant cells	Head and neck	No typical features	Inflammatory stroma with multinucleated non-tumoral giant cells resembling osteoclasts and showing immune-expression of osteoclast/histiocytic markers; higher grade lesion, with moderate to poor differentiation.

**Table 4 ijms-24-11176-t004:** Relative frequency and prognosis of primary cutaneous lymphomas included in the 2018 update of the WHO–EORTC classification [6].

WHO–EORTC Classification 2018	Frequency, %	5-y DSS, %
**Cutaneous T-cell lymphomas (CTCL)**		
Mycosis fungoides (MF)	39	88
Mycosis fungoides variants		
Folliculotropic MF	5	75
Pagetoid reticulosis	<1	100
Granulomatous slack skin	<1	100
Sézary syndrome	2	36
Adult T-cell leukemia/lymphoma	<1	NDA
Primary cutaneous CD30+ lymphoproliferative disorders		
Primary cutaneous anaplastic large lymphoma (C-ALCL)	8	95
Lymphomatoid papulosis (LyP)	12	99
Subcutaneous panniculitis-like T-cell lymphoma	1	87
Extranodal NK/T-cell lymphoma, nasal type	<1	16
Chronic active Epstein–Barr virus (EBV) infection	<1	NDA
Primary cutaneous peripheral T-cell lymphoma, rare subtypes		
Primary cutaneous g/d T-cell lymphoma	<1	11
Primary cutaneous aggressive epidermotropic CD8^+^ cytotoxic T-cell lymphoma (CD81 AECTCL, provisional)	<1	31
Primary cutaneous CD4^+^ small/medium T-cell lymphoproliferative disorder (provisional)	6	100
Primary cutaneous acral CD81 T-cell lymphoma (provisional)	<1	100
Primary cutaneous peripheral T-cell lymphoma, NOS	2	15
**Cutaneous B-cell lymphomas (CBCL)**		
Primary cutaneous marginal zone lymphoma (PCMZL)	9	99
Primary cutaneous follicle center lymphoma (PCFCL)	12	95
Primary cutaneous large B-cell lymphoma, leg type (PCDLBLC, LT)	4	56
EBV1 mucocutaneous ulcer (provisional)	<1	100
Intravascular large B-cell lymphoma	<1	72

CTCL, cutaneous T-cell lymphoma; CBCL, cutaneous B-cell lymphomas; CD8^+^ AECTCL, primary cutaneous aggressive epidermotropic CD8^+^ cytotoxic T-cell lymphoma; DSS, disease-specific survival; NDA, no data available; NOS, not otherwise specified; MF, mycosis fungoides; C-ALCL, primary cutaneous anaplastic large lymphoma; LyP, lymphomatoid papulosis; PCMZL, primary cutaneous marginal zone lymphoma; PCFCL, primary cutaneous follicle center lymphoma; PCDLBLC, LT, primary cutaneous large B-cell lymphoma, leg type.

**Table 5 ijms-24-11176-t005:** Characteristic features of PCFCL with a diffuse growth pattern and PCDLBCL, LT [6].

	PCFCL, Diffuse Large Cell	PCDLBCL, LT
**Clinical presentation**	Localized lesions: head/trunk	Tumoral lesion on lower legs
**Histopathology**		
Tumoral cells	Predominance of large centrocytes; centroblasts may be present	Predominant and confluent sheets of centroblasts and/or immunoblasts
Admixed T cells	Often abundant	Sparse/mainly perivascular
**Immunohistochemistry**		
B-cell lineage markers	CD20+, CD79a+, PAX5+, IgM−, IgD−	CD20+, CD79a+, PAX5+, IgM+, IgD+/−; monotypic light chain expression
Germinal center markers	BCL6+, BCL2−, CD10−	BCL6+, BCL2−, CD10−
Post-germinal center markers	IRF4/MUM1-, FOXP1-	IRF4/MUM1+, FOXP1+
MYC expression	Negative	Positive (65–80%)
CD21/CD35: remnants of FDC networks	Sometimes present	Absent
**Molecular genetics**		
**Gene expression profile**	GCB-type DLBCL	ABC-type DLBCL
**Translocations BCL6, MYC, IgH**	Absent	BCL6 (30%), MYC (35%), IgH (50%)
**Array-based CGH; FISH**	Amplification 2p16.1 Deletion 1p36 Deletion 14q11.2-q12	Deletion 6q arm (BLIMP1:60%) Deletion 9p21.3 (CDKN2A:67%) MYD88
**NF-κB pathway mutations**	No MYD88 mutation	MYD88 (60%), CD79B (20%), CARD11 (10%), TNFAIP3/A20 (40%)

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
