# Peer review of "Advancing Cancer Research: Current Knowledge on Cutaneous Neoplasia"

_ijms, 2023, doi:10.3390/ijms241311176_

Round 1
Reviewer 1 Report
Thank you for the opportunity to review the manuscript.
The topic is important, but I honestly don't understand the selection of certain cutaneous malignancies and the omission of others.
Your title is "Advancing Cancer Research: New Perspectives on Cutaneous Neoplasia" but you have not included some important skin cancers such as:
the majority of adnexal carcinomas (see https://doi.org/10.3390/ijms22105077 https://doi.org/10.3390/ijms22094759)
Merkel cell carcinoma (https://doi.org/10.3390/ijms22126305)
dermatofibrosarcoma protuberans
Kaposi sarcoma
angiosarcoma
cutaneous leiomyosarcoma.
I understand that it could be little progress in rare skin malignancies but it also exists (like imatinib in DFSP).
Therefore, I recommend that you rewrite the article to make it more comprehensive and focus on non-melanocytic skin malignancies. You can leave out malignant melanoma because there are many more new therapies than you have described in the manuscript and it deserves a separate article.
Author Response
Dear Reviewer,
Thank you very much for evaluating our manuscript. Your recommendations have helped us improve our manuscript. Here we address the comments. The changes we have made in the manuscript are highlighted in red.
- Your title is "Advancing Cancer Research: New Perspectives on Cutaneous Neoplasia" but you have not included some important skin cancers such as:
the majority of adnexal carcinomas (see https://doi.org/10.3390/ijms22105077 https://doi.org/10.3390/ijms22094759)
Merkel cell carcinoma (https://doi.org/10.3390/ijms22126305)
dermatofibrosarcoma protuberans
Kaposi sarcoma
angiosarcoma
cutaneous leiomyosarcoma.
I understand that it could be little progress in rare skin malignancies but it also exists (like imatinib in DFSP). Therefore, I recommend that you rewrite the article to make it more comprehensive and focus on non-melanocytic skin malignancies. You can leave out malignant melanoma because there are many more new therapies than you have described in the manuscript and it deserves a separate article
Response: we introduced in our manuscript new data about all the skin cancers mentioned by you above, trying to summarize the latest discoveries in the field of pathology and their treatment. With regard to malignant melanoma, we have reviewed the information in the manuscript and added new data.
Thank you again for reviewing our manuscript
Reviewer 2 Report
The review by Laura Stătescu aims to summarize important changes in the approach to melanoma, non-melanoma skin cancers, and cutaneous T and B cell lymphomas. Overall, a multidisciplinary approach involving dermatology, medical oncology, and radiation oncology is necessary for optimal patient management of all types of skin cancers.
I have the following comments/concerns about this manuscript:
1. Some sentences are presented vaguely without providing further details. Some information provided is also incorrect. I recommend the authors avoid generalised statements and where relevant, provide specific details about their arguments. The presentation of the manuscript needs to be improved.
Some examples:
Line 68: The subheading does not seem to be related to diagnosis and diagnostic approaches, but the paragraph under the subheading mainly discusses the diagnosis and diagnostic approaches. For Risk factors, only those related to genetic mutations have been discussed, which are not the only risk factors for melanoma development.
Line 72: “Melanoma, the 3rd most common skin cancer, is a malignant melanocytic tumour, classified into 4 major histological subtypes”: The subtypes need to be stated.
Line 75: “Furthermore, among young adults aged 15 to 29 residing 75 in the US, cutaneous neoplasia ranks as the second most frequently diagnosed form of cancer.” I believe it is cutaneous melanoma, NOT cutaneous neoplasia.
Line 95-98: Doesn’t make any sense, and is disorganised, misplaced.
Line 100: Reflectance confocal microscopy (RCM) is already spelled out in the previous paragraphs.
Line 102: “While RCM is widely utilized in Europe, it has started to become accessible in select centres across the United States”: needs to be supported by the literature. There are also numerous other instances that need to be supported by literature.
Line 194 and 242: better to be summarised in tables.
Line 264: these sub-headings are very confusing and need to be re-arranged. I do not see why there is a sub-heading for BCC in line 245, and one here.
Other comments:
2. The title is not quite related to the content. When the authors state “New Perspectives on Cutaneous Neoplasia”, one would expect the content to critically review the existing knowledge and literature and provide or suggest “new perspectives”. In its present form, the content mainly summarises our current understanding of the cutaneous neoplasia, with minimal to no critique or without providing any “new perspective”.
3. There are imbalances and inconsistencies in covering different aspects of different cutaneous neoplasia. As an example, histological subtypes of CSCC and BCC are provided in great detail, while in Melanoma, the genetic subtype is the focus of this paper. The same with many other topics that are covered for each lesion subtype.
4. Also, the manuscript requires extensive proofreading and a better presentation of the content. There are some grammatical errors, and some sentences require re-phrasing.
The manuscript requires extensive proofreading and a better presentation of the content. There are some grammatical errors, and some sentences require re-phrasing.
Author Response
Dear Reviewer,
Thank you very much for evaluating our manuscript. Your recommendations and comments have helped us improve our manuscript. Here we provide the requested corrections and address the comments. The changes we have made in the manuscript are highlighted in red.
1.Some sentences are presented vaguely without providing further details. Some information provided is also incorrect. I recommend the authors avoid generalised statements and where relevant, provide specific details about their arguments. The presentation of the manuscript needs to be improved.
Response: We reformulated some phrases, re-organized them and reviewed English language.
Line 68: The subheading does not seem to be related to diagnosis and diagnostic approaches, but the paragraph under the subheading mainly discusses the diagnosis and diagnostic approaches. For Risk factors, only those related to genetic mutations have been discussed, which are not the only risk factors for melanoma development.
Response: Thank you for the suggestion; we considered to mention only the genetic mutations due to their importance in the staging and choosing of treatment options.
Line 72: “Melanoma, the 3rd most common skin cancer, is a malignant melanocytic tumour, classified into 4 major histological subtypes”: The subtypes need to be stated.
Response: We added the histological subtypes.
Line 75: “Furthermore, among young adults aged 15 to 29 residing 75 in the US, cutaneous neoplasia ranks as the second most frequently diagnosed form of cancer.” I believe it is cutaneous melanoma, NOT cutaneous neoplasia.
Response: We corrected
Line 95-98: Doesn’t make any sense, and is disorganised, misplaced.
Response: We reformulated
Line 100: Reflectance confocal microscopy (RCM) is already spelled out in the previous paragraphs.
Response: We corrected
Line 102: “While RCM is widely utilized in Europe, it has started to become accessible in select centres across the United States”: needs to be supported by the literature. There are also numerous other instances that need to be supported by literature.
Response: We reformulated
Line 194 and 242: better to be summarised in tables.
Response: We summarized the information in table 1
Line 264: these sub-headings are very confusing and need to be re-arranged. I do not see why there is a sub-heading for BCC in line 245, and one here.
Response: We re-arranged the paragraphs
Other comments:
2.The title is not quite related to the content. When the authors state “New Perspectives on Cutaneous Neoplasia”, one would expect the content to critically review the existing knowledge and literature and provide or suggest “new perspectives”. In its present form, the content mainly summarises our current understanding of the cutaneous neoplasia, with minimal to no critique or without providing any “new perspective”.
Response: We changed the title of the manuscript, replacing "New perspetives" with "Current knowledge". At the suggestion of another reviewer, we have also added more up-to-date information on 6 other types of skin cancer.
3.There are imbalances and inconsistencies in covering different aspects of different cutaneous neoplasia. As an example, histological subtypes of CSCC and BCC are provided in great detail, while in Melanoma, the genetic subtype is the focus of this paper. The same with many other topics that are covered for each lesion subtype.
Response: We tried to improve this aspect adding more information. Besides, in the case of melanoma, the genetic subtype is very important in the staging of the disease and choosing the right treatment (targeted therapy / immunotherapy), but is also important for the prognosis (as the latest scientific papers mentioned – ex. DecisionDX Melanoma)
4.Also, the manuscript requires extensive proofreading and a better presentation of the content. There are some grammatical errors, and some sentences require re-phrasing.
Response: We tried to correct.
Thank you again for reviewing our manuscript,
Reviewer 3 Report
While this manuscript is presenting substantial information, the first section (first three pages covering metastatic melanomas) is so disorganized that it was quite laborious to read through and offer appropriate revisions. Once past that first third of the manuscript, the submission is more organized and cohesive, although I do feel that basal cell carcinoma nevoid syndrome should at least be mentioned for completeness.

Author Response
Dear Reviewer,
Thank you very much for evaluating our manuscript. Your recommendations and comments have helped us improve our manuscript. Here we provide the requested corrections and address the comments. The changes we have made in the manuscript are highlighted in red.
This submission reviews the primary types of abnormal skin growths and their standard treatments. Review of the English language is needed, mainly writing sentences directly rather than in passive tones. The lack of references in specific sections and the mix of overview statement and specific focus within paragraphs lead to confusion. Re-organization would enhance the clarity and impact of this submission.
Response: We reformulated some phrases, re-organized them and reviewed English language.
Abstract - The sentence starting with “SeC requires an early treatment being….” needs to be written as a more direct statement.
Response: We reformulated
Introduction – The introduction references only WHO-EORTC while providing a general overview of the manuscript focus. References are needed.
Response: We added references
Malignant melanoma (MM) – The authors start the review of cutaneous neoplasia with the most lethal type. However, in the first paragraph of this section, the authors use the term “cutaneous neoplasia” making it unclear when the authors are stating incidence rates for MM or all types of cutaneous cancers. As the estimated incidence of basal cell carcinoma alone exceeds the annual incidence of all other cancer types in the U.S., the statement that “…cutaneous neoplasia ranks as the second most frequently diagnosed form of cancer” is not appropriate. Testicular, thyroid, and breast cancer are more common than melanoma among 15 – 29 year olds in the U.S according to the National Cancer Institute.
Response: We Corrected; it was a mistake: cutaneous neoplasia – cutaneous melanoma
In the next paragraph, the authors compare MM rates per 100,000 across a handful of Nations. Focusing on increased incidence (Internationally and specifics) in the first MM paragraph and then the comparison of rates per 100,000 in the second paragraph would be more cohesive.
Response: We Corrected the first paragraph, it was a mistake: cutaneous neoplasia – cutaneous melanoma; now is a logical course of the text
The third paragraph provides the basics of clinical MM recognition without any references. The “Ugly Duckling” sign is a colloquialism and needs a brief explanation in parenthesis such as (moles that are different or stand out in comparison to other moles on one’s body). References are needed as the various diagnostic techniques are discussed. Likewise, the frequency of use of these approaches (next paragraph) and proportionate diagnoses by technology approach (4th paragraph in the section) need citations for substantiation.
Response: We added references and information
The 5th paragraph of the MM section starts with a sentence fragment and this paragraph needs a clear focus as to what it is trying to present.
Response: We added references and information
The information being presenting on page 4 starting with Management of Local/Locoregional Disease would be better summarized in a table format.
Response: We summarized the information in table 1
The discussion of BCC is more cohesive. As the authors are aiming for a comprehensive review of skin lesions, basal cell nevus syndrome (also called nevoid basal cell carcinoma syndrome or Gorlin syndrome) should at least be mentioned with a very brief note that diagnostic routines may need to expedited and radiation avoided among afflicted individuals.
Response – We added information on Gorlin syndrome
It is unclear why the second subtitle is “BCC, basal cell carcinoma; SCC, squamous cell carcinoma when the authors are still discussing only BCC in this section. This mixed sub-heading should be deleted and then the next section of 2.2.2 should be subtitled SCC, squamous cell carcinoma.
Response : We corrected and re-arranged the subtitles
The discussion of Sebaceous carcinoma is well-focused and follows an appropriate path of incidence, diagnoses, tumor characteristics and DNA alterations, ending with treatment options. 2 Similarly, the discussions of primary cutaneous lymphomas, cutaneous T-cell lymphomas, and cutaneous B-cell lymphomas can be well-followed. This manuscript provides a comprehensive review of skin lesions but currently lacks a cohesive presentation and sufficiently detail foundations for many statements, particularly in the metastatic melanoma review (first three pages of the manuscript). Statements need to have supporting references. With a more focused, readable presentation, this review will be a valuable contribution to the literature.
Response: Thank you again for reviewing our manuscript.
Reviewer 4 Report
Please see in the attachment.

Author Response
Dear Reviewer,
Thank you very much for evaluating our manuscript. Your recommendations and comments have helped us improve our manuscript. Here we provide the requested corrections and address the comments. The changes we have made in the manuscript are highlighted in red.
Dear Author, I believe your review is ambitious and worthy, but from my view for this high impact journal some changes are needed.
1.Melanoma, line 71 to 91: personally, I feel this text is quite simple, no novelties are expressed in this part. In an advance review this information is supposedly know by the readers.
Response: Thank you very much for your remark, you are right; we chose to present at the beginning a few data about the subject, trying to explain the importance of the data discussed in the next paragraphs, where we added new information
2. Line 100: Confocal it is not a diagnosis approach that eliminate a skin biopsy, that is a wrong statement, please correct it, is important. The gold standard is always the histological examination.
Response: We reformulated and explained that the gold standard remain the histological examination.
3. Line 120: Need a reference, what test is this and in which work has been validated?
Response: We Introduced the reference no 6, the test is presented in concordance with the guidelines presented on the website of the American Cancer Society
4. Line 156: novel indices, as PD-L1, for example, authors should go deeper, what is the transcendence?
Response: We reformulated and completed with new informations
5.Line 178: Changes with respect to? Previous situation? Changes are based in and improve what?
Response: We reformulated – changes in the tumor, nodal and metastasis staging.
6. Line 245: Again, very simple introduction for an advanced journal. This is not new. Table I: does not enough quality for this journal, it is for other profile or readers. Line 264: Nothing new in this paragraph.
Response: Dear reviewer, you are right; at the suggestion of another reviewer we changed the title of the manuscript replacing the phrase "New perspectives" with "Current knowledge"; so we decided to leave the paragraphs with general information as well. We have added more information and news in the case of 6 other tumors.
7.Line 356. This is the typical immunohistochemistry of an SCC, but it is feasible?. Do we used it normally in clinical practice? It has a future? Authors are only describing without an analysis or a discussion.
Response: Thank you for your comment; we added information and discuss on this topic.
9. Line 421: Why do you include Sebaceous carcinoma and not Merkel carcinoma or fusocellular carcinomas or basal squamous carcinoma? Why do you select this one?
Response: At your suggestion and that of another reviewer, we have added more up-to-date information on 6 other types of skin cancer.
Table 3: it is copy-pasting the EORTC classification without a deeper valoration or discussion of the authors.
Response: In this table we added information on primary cutaneous lymphomas included in the 2018 update of the WHO-EORTC classification
My conclusions are that the authors are trying to reach a very high goal, that is trying to summarize all the advances and novelties of skin cancer, all types. From my view, these kind of important goal should be approached with the supposing that the readers know the basis and go directly to the novelties.
Response: Thank you again for your effort reviewing our paper.
Round 2
Reviewer 1 Report
Thank you for implementing my comments. The manuscript has improved significantly. In my opinion, now it's suitable for publication.
Author Response
Dear Reviewer,
Thank you for reviewing our manuscript.
Reviewer 3 Report
IMJS – 2440136_V2 Advancing Cancer Research: Current Knowledge on Cutaneous Neoplasia
This submission reviews the primary types of abnormal skin growths and their standard treatments.
Review of the English language is needed, mainly writing sentences directly rather than in passive tones. The lack of references in specific sections and the mix of overview statement and specific focus within paragraphs lead to confusion. Re-organization would enhance the clarity and impact of this submission.
Abstract – The authors use “extention” in the Abstract and a few times in the manuscript. The correct word is “extension”
Introduction – The authors have added references.
Malignant melanoma (MM) – The authors quote an author (ref. #2) that is quoting a review of SEER information from 1975 – 2000 (https://academic.oup.com/oncolo/article/11/6/590/6398200 ). The interpretation of the original article is not entirely correct with the summation, and is no longer accurate according to recent national statistics. What is correct is that melanoma incidence rates have continued to increase among the younger age groups (15 – 29 year olds – mostly due to the 20 – 29 year old age groups) and melanoma is the fourth most frequently diagnosed invasive cancer in the age group (https://www.cancer.gov/types/aya ; National Cancer Institute). Please correct the information rather than continue to spread inaccurate information.
Table 1 is a much-improved approach to presenting the Management of Local/Locoregional MM. Nicely done.
The paragraph that begins “In order to distinguish (no “ing”) SCC from …..” needs appropriate citations for the first five statements. Similarly, the 2.2.3 section Cutaneous malignant adnexal neoplasms, needs references for the first three statements and the final paragraph prior to the Management section and the final paragraph of the Management section. Startng on page 18 with the treatment of cutaneous leiomyosacomas (i.e. “Regarding the treatment….”), references are needed. Page 19 third paragraph “The most important prognostic factors…” – references are needed.
This significantly improved manuscript provides a comprehensive review of skin lesions. Minor revision is needed to correct mis-information and provide supporting references in the few areas in which these are lacking. With these revisions, this review will be a valuable contribution to the literature.
Minor revisions
Author Response
Dear Reviewer,
Thank you so much for re-evaluating our manuscript. We took into account all your suggestions and made the necessary changes and additions. Also, we added eight new references in the sections where they were missing.
Reviewer 4 Report
Dear authors,
You have made so many changes in your manuscript with a great effort. Actually, the manuscript has improved greatly.
Best wishes,
English is ok from my view
Author Response
Dear Reviewer,
Thank you again for reviewing our manuscript.